# The Red Sea Deep Water is a potent source of atmospheric ethane and propane

E. Bourtsoukidis [1*], A. Pozzer[1], T. Sattler[1], V.N. Matthaios[2], L. Ernle[1], A. Edtbauer[1], H. Fischer[1], T. Könemann[3], S. Osipov[1], J.-D. Paris[4], E.Y. Pfannerstill [1], C. Stönner[1], I. Tadic [1], D. Walter [3,5], N. Wang[1], J. Lelieveld [1,6] & J. Williams[1,6]

Non-methane hydrocarbons (NMHCs) such as ethane and propane are significant atmospheric pollutants and precursors of tropospheric ozone, while the Middle East is a global emission hotspot due to  extensive oil and gas production. Here we compare in situ hydrocarbon measurements, performed around the Arabian Peninsula, with global model simulations that include current emission inventories (EDGAR) and state-of-the-art atmospheric circulation and chemistry mechanisms (EMAC model). While measurements of high mixing ratios over the Arabian Gulf are adequately simulated, strong underprediction by the model was found over the northern Red Sea. By examining the individual sources in the model and by utilizing air mass back-trajectory investigations and Positive Matrix Factorization (PMF) analysis, we deduce that Red Sea Deep Water (RSDW) is an unexpected, potent source of atmospheric NMHCs. This overlooked underwater source is comparable with total anthropogenic emissions from entire Middle Eastern countries, and significantly impacts the regional atmospheric chemistry.

[1] Department of Atmospheric Chemistry, Max Planck Institute for Chemistry, Mainz 55128, Germany. [2] School of Geography, Earth and Environmental Sciences, University of Birmingham, Edgbaston, Birmingham B15 2TT, UK. [3] Department of Multiphase Chemistry, Max Planck Institute for Chemistry, Mainz 55128, Germany. [4] Laboratoire des Sciences du Climat et de l'Environnement, CEA-CNRS-UVSQ, UMR8212, IPSL, Gif-Sur-Yvette, France. [5] Max Planck Institute for Biogeochemistry, Hans-Knöll-Straße 10, 07745 Jena, Germany. [6] Energy, Environment and Water Research Center, The Cyprus Institute, Nicosia, Cyprus. *email: e.bourtsoukidis@mpic.de

The Middle East accommodates more than half of the world's known oil and gas reserves[1]. Fossil fuel exploitation in this region is responsible for the release of large amounts of gaseous pollutants into the atmosphere, including methane ($CH_4$)[2] and non-methane hydrocarbons (NMHCs)[3]. Ethane and propane have the strongest sources[4], and being relatively long-lived (ethane ca. 2 months, propane ca. 2 weeks)[5] are ubiquitous in the global atmosphere. Atmospheric oxidation of NMHCs in the presence of nitrogen oxides ($NO_x$) leads to production of tropospheric ozone[6,7] and peroxyacetyl nitrates (PAN) that are phytotoxic[8–10] and harmful to human health[11]. The abundance of NMHCs and $NO_x$ in combination with the intense photochemistry in the Arabian Basin results in extremely high ozone mixing ratios that can reach up to 200 ppb[12].

Globally, the atmospheric concentrations of ethane and propane exhibit temporal trends that are closely related to anthropogenic activities. The general decline in fossil fuel emissions toward the end of the twentieth century resulted in a decline of global atmospheric ethane and propane[13]. Conversely, the subsequent expansion of US oil and natural gas production has led to a reversal of their global atmospheric trends, with emissions increasing since 2010[3].

While anthropogenic activities substantially influence the emission rate and composition of atmospheric hydrocarbons, Earth's natural degassing is also a significant source[14]. Natural geologic (i.e., mud volcanoes, onshore and marine seeps, and micro seepage, geothermal and volcanic) sources contribute to both ambient ethane and propane concentrations, and their inclusion in global emission inventories helps to better explain the reported values from the expanding global observation network[15]. Indeed such sources will have dominated preindustrial emissions.

During the AQABA ship campaign, which took place between July and August 2017, NMHCs were monitored around the Arabian Peninsula (Supplementary Fig. 1). By comparing the observations with model simulations, we aim to evaluate the emission inventories and atmospheric chemistry mechanisms while focusing on the most abundant anthropogenic hydrocarbons: ethane and propane. The largest measurement/model discrepancy was observed over the northern part of the Red Sea, which was investigated in terms of possible underestimation of existing sources and emission patterns (i.e., ratios between the measured hydrocarbons) that are derived by using positive matrix factorization analysis.

## Results and discussion

**Observations and model simulations**. The atmospheric mixing ratios of ethane and propane ranged over three orders of magnitude around the periphery of the Arabian Peninsula (Supplementary Figs. 2–4). From their tropospheric background values over the Arabian Sea (defined as the lowest 10% of data points: $0.18 \pm 0.02$ ppb for ethane and $0.02 \pm 0.01$ ppb for propane), to a maximum of ~50 ppb over the Arabian Gulf, variations in absolute and relative abundance of the NMHCs indicated multiple emission sources[16,17]. The high mixing ratios over the Arabian Gulf and Suez Canal could be attributed to emissions from the intense oil and gas activities and urban centers, respectively[16]. However, in the region with the second highest average abundance, namely the northern part of the Red Sea, the levels of the measured NMHCs could not be attributed to a known source[16].

To identify the source and to generally evaluate the Middle Eastern NMHC emission patterns, a state-of-the-art atmospheric chemistry model (EMAC, ECHAM5/MESSy for Atmospheric Chemistry)[18,19] was used to simulate the concentrations along the route. As emission inventory input, we used the most recent version of the Emission Database for Global Atmospheric Research (EDGAR v4.3.2)[20,21], which was further upgraded to include gas flare emissions and geothermal sources (see the "Methods" section).

The observed ethane mixing ratios were reproduced by the model for most of the route (Fig. 1), indicating that emission sources and atmospheric processes in the Middle East region are generally well understood. Significant model underestimations (Suez Canal, northern Arabian Gulf) and overestimations (Gulf of Oman) occurred for short periods only during the first leg of the route, suggesting local, small-scale inconsistencies in the emission sources. The only region that was inadequately simulated during both legs of the route was the northern part of the Red Sea where measured mixing ratios of ethane and propane were up to about 20 (average ± standard deviation = $4.3 \pm 3.8$) and 40 (average ± standard deviation = $7.8 \pm 5.9$) times higher than model predictions. According to the model results, biomass burning, fuel production and transmission, and transformation industry emissions regulate the regional hydrocarbon abundance (Fig. 1d; Supplementary Fig. 5). However, neither the dominant nor any of the 15 inventory sources was able to explain the observations, even when the emission strength was varied (Supplementary Fig. 6).

**Source apportionment**. To derive the hydrocarbon signature of the potent unidentified source, a well-established receptor model (positive matrix factorization; US EPA PMF 5.0) was utilized[22]. Receptor models use ambient observations to apportion the observed species concentrations to signature sources by assessing changes in species correlation with time and finding the optimum solution that explains the concentrations of all observed constituents.

The PMF analysis of the northern Red Sea data identified 4 distinct emission sources/factors (Fig. 2a). Factor 1 illustrates an emission source that is rich in C2–C6 hydrocarbons, alkenes and acetonitrile (a biomass-burning tracer[23]), with contributions mainly during the first leg when the air masses originate from the Suez Canal (Supplementary Fig. 7). In addition, it correlates with anthropogenic activity markers such as acetone, methanol, and acetaldehyde (Supplementary Fig. 8), confirming the urban character of this emission source. By contrast, factor 4 was significant only during the second leg (Supplementary Fig. 9) when the air originated from the Sinai Peninsula. The small concentration contribution and back-trajectory-specific direction suggest a background signature from a region with small emission sources. Factor 3 apportions marine traffic emissions that are clearly distinguishable from the sporadic occurrences (Supplementary Fig. 10), high ethene, large alkanes (>C4), and the absence of ethane[16] in the emission pattern.

The remaining factor 2 is characterized by exceptionally high alkane concentrations that decrease with increasing carbon number. Alkene contribution is negligible, and in combination with the absence of acetonitrile in the emission signature, factor 2 points toward a non-anthropogenic emission source. The source contribution to the measured signal is expressed by the significance of the factor that is termed as factor strength. As shown in Fig. 2b, this source contributed most to the measurements over the northmost part of Red Sea and in particular between 23° and 27° latitude. Since the model underestimation (expressed by the ratio between measurements and models) increases with the strength of factor 2 (Fig. 2c), it becomes evident that it represents the missing source.

Summarizing, the high ethane and propane mixing ratios that were observed over the northern part of Red Sea could not be

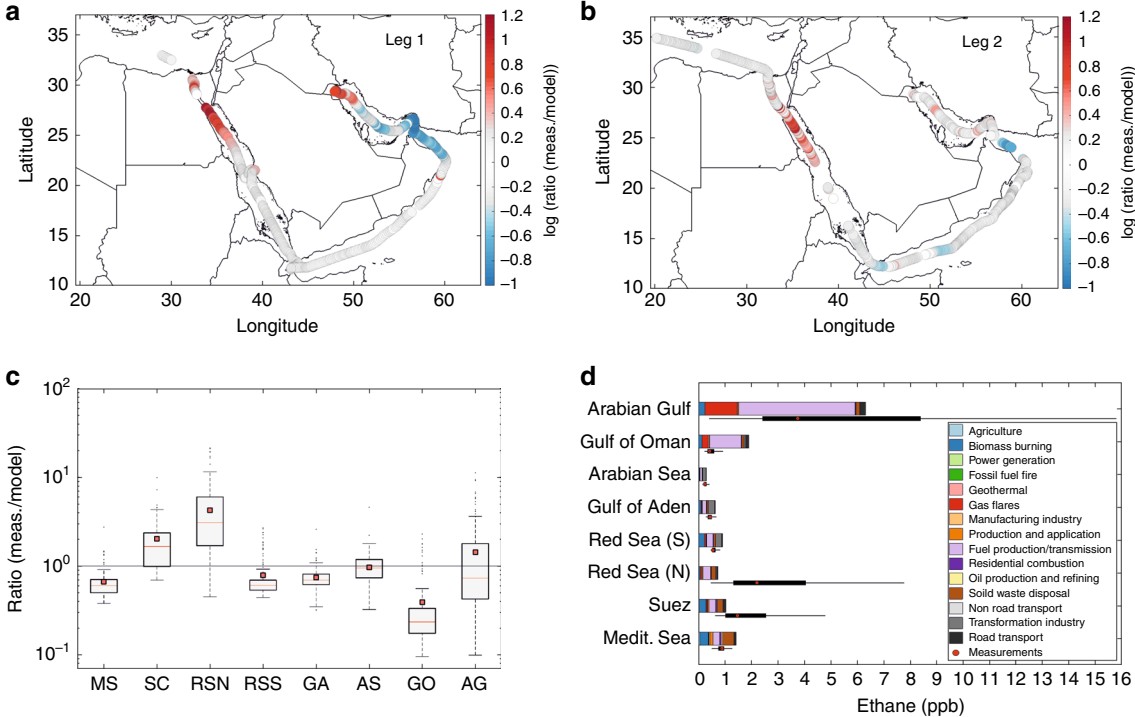

**Fig. 1 Comparison between ethane measurements and model simulations.** Timeline ratios are displayed for both leg 1 (**a**) and leg 2 (**b**). In **c**, geospatial ratio statistics are displayed with the boxplots that illustrate the median with red line and the mean with red squares. The bottom and top edges of the box indicate the 25th (q1) and 75th (q3) percentiles, respectively. The boxplot draws points as outliers if they are greater than $q3 + w \times (q3 - q1)$ or less than $q1 - w \times (q3 - q1)$. The whiskers correspond to $\pm 2.7\sigma$ and 99.3% coverage if the data are normally distributed. In **d**, geospatial measured volume mixing ratios of ethane are shown with black boxplots where the red circles are the median measured values and the whiskers are defined as in (**c**). The modeled mixing ratios are in bars for each emission sector.

explained by the known sources that are included in the emission inventory (Supplementary Fig. 6). PMF analysis identified the emission source signature, suggesting that it is distinct from other known sources, of non-anthropogenic origin, and specific to the region of the Red Sea. Back-trajectory calculations (Supplementary Fig. 7) show that the origin of the air masses remained unchanged along each leg. The highest ethane mixing ratio underprediction occurred over two hot spots pointing toward a local source. One possibility is that the missing source of hydrocarbons is the sea. Marine emissions of NMHCs have been documented previously[24–26], however, fluxes were low. If the hydrocarbons originate from the sea, then an exponential relationship between the measured mixing ratios and wind speed can be expected due to the flux dependency on wind strength[27,28]. Indeed, ethane, propane, butanes, and methane mixing ratios do display exponential increases with the wind speed that additionally correlates with the model underestimation (Supplementary Fig. 11). Methane in particular was substantially increased over the northern Red Sea, with an enrichment (i.e., subtracted backgrounds) methane to ethane ratios of $93 \pm 77$, considerably higher than the respective ratios observed over the Arabian Gulf ($35 \pm 23$) that represents the high end of oil- and gas-processing-related ratios[3].

**Red Sea Deep Water**. The Red Sea lies between the Arabian and African continental plates and has some unique geological features. The southern Red Sea floor has been spreading for the past 5 million years, while the northern part is in a stage of continental rifting[29]. Distinct movements of the tectonic plates have led to a partly fractured sea floor and the formation of numerous brine-

filled pools that are characterized by close-to-solubility limit halite (mineral form of NaCl) concentrations and strong temperature gradients[30,31]. Generally, the water occupying depths from 300 to 2000 m in the Red Sea is recognized as the warmest and saltiest deep water in the world with pronounced seasonality[32], although the rates and mechanisms of its renewal remain uncertain.

Hydrocarbon release from the Red Sea floor can occur through direct fluid seepage from hydrocarbon reservoirs deposited above offshore rocks, located between 25 and 28°N (e.g., Rudeis and Kareen formation). In addition, the Gulfs of Suez and Aqaba contribute to the RSDW through bottom-trapped density outflows[33]. Considering that this region is known for the large oil and gas reserves, natural seepage and crude oil/gas seepage from leaky subsea wells could be a significant submarine source of NMHCs in this region. Finally, the numerous brine pools that are located on the sea floor need to be considered. Depending on their chemical composition, the brine pools are classified into two distinct types[31]. The formation of Type I brine pools is controlled by evaporate dissolution and sediment alteration, characterized by exceptionally high methane and hydrocarbon concentrations. Short-chained hydrocarbons are formed by the degradation of long-chained hydrocarbons that originate from the organic-rich sedimentary rocks[34] and the bioproduction in the brine's water to sediment interface[35,36]. Type II brine pools are controlled by volcanic/magmatic alterations and are poor in organic material. They are more common and are frequently found in the southern and middle part of the Red Sea floor. In contrast, only two brine pools are classified as type I: the *Oceanographer* deep (26°17.2′N, 35°01.0′E)[37] and the *Kebrit* deep (24°43.1′N, 36°17′E)[38]. *Oceanographer* in particular is known to contain high methane

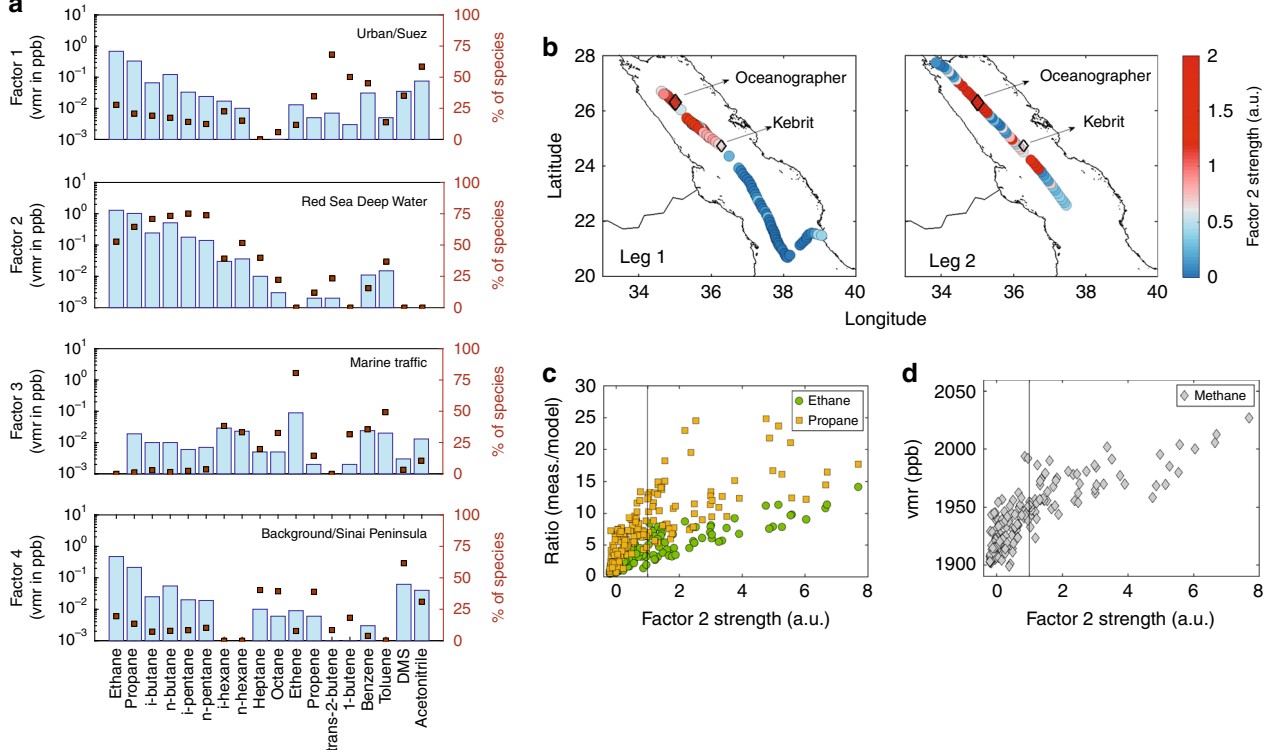

**Fig. 2 Source apportionment for the northern Red Sea region. a** Profiles derived from positive matrix factorization (PMF) analysis. The blue bars indicate the volume mixing ratio contribution from each source and the brown squares the % contribution of each species to the respective factor (sum of factors = 100%). In **b**, factor 2 strength (average strength = 1) timelines are illustrated for both legs. In **c**, factor 2 strength is correlated with the ratio between measurements and models for ethane and propane, and in **d** with the measured methane. Further explanation on the source assignment for each factor is shown in Supplementary Figs. S8–S10.

concentrations (4921 µl/l[39] (average methane inside type II brines is 0.04 ± 0.04 µl/l))[31] and it is located directly underneath the region with the highest discrepancy between NMHC measurements and model simulations. The mixing between the dense brine pools and RSDW occurs via diffusion across the strong salinity gradient, and the reported methane emissions are relatively weak with values up to 393 kg/yr for the Kebrit deep[40]. However, the impact of above-brine water currents that may enhance the fluxes has not been quantified, so this emission estimate remains uncertain.

Considering the aforementioned potential sources, it seems entirely plausible that the RSDW is highly enriched in ethane and propane from a deep sea source. The transfer from deep water to the surface can be relatively rapid due to the exceptionally effective vertical mixing of the Red Sea deep water and the outflows from the Gulfs of Suez and Aqaba that have been considered to be important for the RSDW renewal in the period 1982–2001[32,41,42].

The upwelling of the intermediate and deep water takes place in the narrow band along the Egyptian coast. Spatially, the upwelling is restricted to the northmost Red Sea (north of 24°N) and coincides with the location of the missing NMHC source. The complex overturning circulation in the Red Sea has a pronounced seasonal cycle[41,43] that modulates the vertical transport and can potentially amplify the emissions to the atmosphere during winter. The upwelling is weaker in summer compared with the winter since atmospheric cooling drives the open water convection and enhances the vertical mixing in the water column. Due to the weak stratification of the water column in the northern part of the basin, the convective mixing can be especially deep and reaches the sea floor[42,44,45]. Model estimates of the renewal times range from 19 to

90 years, while tracer studies indicate somewhat faster renewal times to about 26 years[33,42]. Further, mesoscale eddies are particularly effective in the areas of the *Oceanographer* and *Kebrit* deeps[46] and may contribute to the vertical transport of hydrocarbons. Eddies may also affect deep-water environment with downward effective transfer rate of 200–600 m day⁻¹ as measured in the Pacific Ocean[47].

While the relative significance of the various submarine hydrocarbon sources cannot be ascertained, we assume that their cumulative contribution represents the missing source derived in this study. This assumption is supported by the similarity in the PMF-derived chemical emission profiles (increased alkane concentrations and the absence of alkenes and other anthropogenic tracers in the emission signature). Therefore, we surmise that methane and non-methane hydrocarbons can potentially reach the surface and degas into the atmosphere following air–sea exchange mechanisms.

**Flux calculations.** To test this hypothesis, two source points (over two model resolution grids (1.1 × 1.1°) with intensity 2:1 from north to south) were added to the model simulation as additional point sources from the ocean surface at the location of the type I brine pools. While Type I brine pools were chosen as the reference location, the emission points cover a large area and thereby include emissions from all aforementioned potential sources. Initially, approximate emission rates that match the factor 2 signature were imported. The measurement to model ratio output for NMHCs was substantially improved with median values deviating from unity by only ca. ±30%. Therefore, the emission rates were fine-tuned so that the measurement/model median ratio was

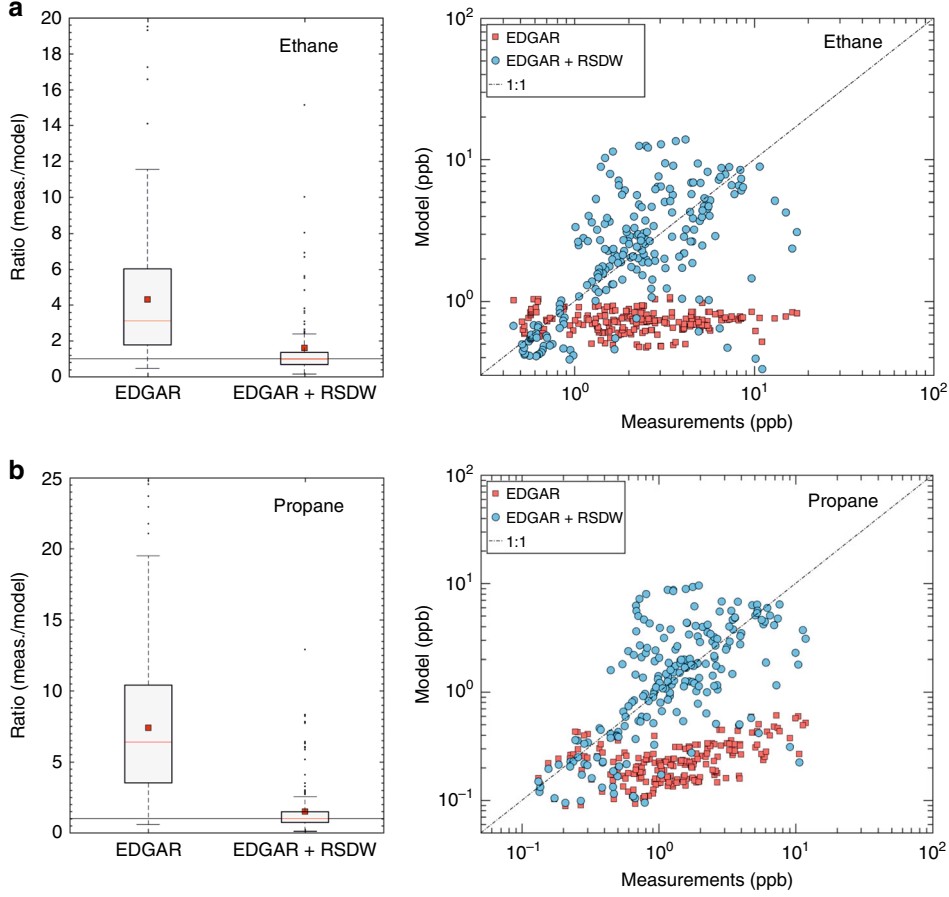

**Fig. 3 Model simulations and measurements over the northern Red Sea.** Ethane (**a**) and propane (**b**) relationships are illustrated for the EDGAR v4.3.2 inventory and compared with the addition of the Red Sea Deep Water (RSDW) emissions in the inventory input. The boxplots illustrate the median with red line and the mean with red squares. The bottom and top edges of the box indicate the 25th (q1) and 75th (q3) percentiles, respectively. The boxplot draws points as outliers if they are greater than $q3 + w \times (q3 − q1)$ or less than $q1 − w \times (q3 − q1)$. The whiskers correspond to $\pm 2.7\sigma$ and 99.3% coverage if the data are normally distributed.

equal to 1, and hence the emission strength of the RSDW could be ascertained. This resulted in predicted emission rates for ethane $(0.12 \pm 0.06 \, \mathrm{Tg \, yr^{-1}})$, propane $(0.12 \pm 0.06 \, \mathrm{Tg \, yr^{-1}})$, i-butane = $0.03 \pm 0.03 \, \mathrm{Tg \, yr^{-1}}$, and n-butane = $0.07 \pm 0.06 \, \mathrm{Tg \, yr^{-1}}$ (uncertainties are based on the standard deviation of the measurement/model ratio values within the 25th and 75th percentile). Increasing uncertainties will be introduced when considering the seasonality of the deep-water circulation, the wind speeds at the air–sea interface, and the potential temporal variability of the emissions (i.e., triggered events by the increased seismic activity in the region[48]). As a final step, the derived emission rates were added to the model and the simulations were repeated. The inclusion of the RSDW emissions in the emission inventory significantly improves the model-measurement comparison, making it equivalent to the agreement seen elsewhere on the route (Fig. 3; Supplementary Fig. 12, Supplementary Fig. 6q). The uncertainties here are likely associated with water current circulation with depth and the exact location of the degassing points. Furthermore, the seabed emissions are likely higher than those reported due to oxidation and bacterial degraders in the water column[49,50].

Considering the linearity between the measured ethane and methane mixing ratios (Supplementary Fig. 13) and by assuming common source origin, an emission rate of ca. 1.3 Tg CH$_4$ yr$^{-1}$ is derived. While this rate is only a small fraction of the global natural methane sources $(238–484 \, \mathrm{Tg \, CH_4 \, yr^{-1}})$[51], it is

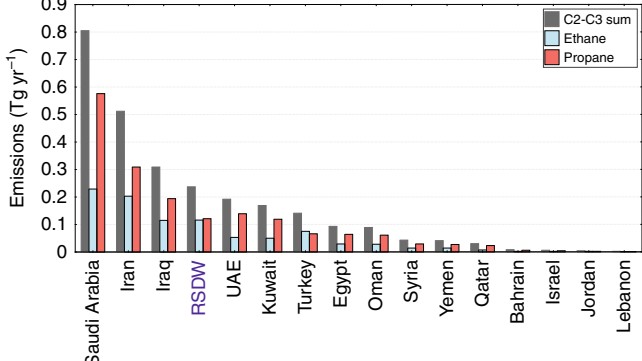

**Fig. 4 Emissions of ethane and propane.** The emissions from Middle Eastern countries compared with the emissions from the northern Red Sea Deep Water (RSDW).

responsible for the high ambient methane mixing ratios observed over the northern Red Sea (average = $1.94 \pm 0.03$ ppm).

**Implications**. The degassing rates for ethane and propane derived here are considerable and comparable in magnitude with the

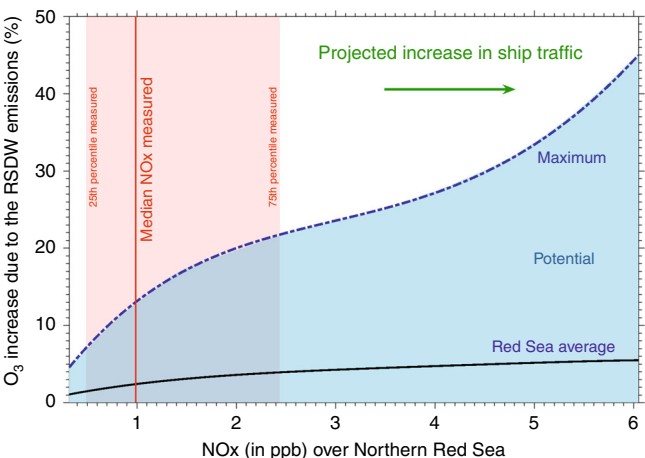

**Fig. 5 Projected implications on O₃ formation.** Increase in O₃ abundance over the Northern Red Sea due to the Red Sea Deep Water (RSDW) emissions. **a** Series of ambient NO$_x$ mixing ratios was simulated with the EMAC model by modifying the NO$_x$ emissions due to shipping. The red line indicates the median NO$_x$ mixing ratios over the Northern Red Sea for both legs. Note that over the northern Red Sea the campaign average of NO$_x$ ±s.d. mixing ratio was measured to be 2.75 ± 6.29 ppb.

emissions from several Middle Eastern countries, known to be exceptional sources related to the hydrocarbon industry (Fig. 4). The combined ethane and propane emission rates rival those from countries, with the most intense oil and gas activities, such as the United Arabic Emirates (UAE), Kuwait, and Oman. In addition, the associated methane emissions from the RSDW are a hitherto unaccounted source of atmospheric methane. Considering the seasonality of the deep-water circulation, it is likely that the emissions to the atmosphere will be further enhanced during the wintertime.

While in much of the Middle East NO$_x$ abundance is a rate-limiting factor in oxidant photochemistry[12], it could be expected that over the Red Sea NMHCs are rate limiting due to small upwind anthropogenic sources. To evaluate the implications of the newly discovered NMHC emission source for atmospheric chemistry, the differences in key atmospheric constituents (hydroxyl radical (OH), ozone (O₃), and PAN) were investigated for the entire area of Northern Red Sea using the model with and without the inclusion of the RSDW sources of ethane, propane, and butanes. Interestingly, summertime average OH depletion is significant over the degassing spots (≈−40%; max = −70%; Supplementary Fig. 14). Downwind of the source location, summertime average ozone production is somewhat enhanced (up to 11%; Supplementary Fig. 15) while there is a prodigious increase in PAN abundance (≈+102%; max = 750%; Supplementary Fig. 16). This represents a significant deterioration of regional air quality, as PAN, a lachrymator and urban smog component is harmful to human health[52,53], and directly related to ethane concentrations[54].

In the coming decades, ship traffic through the Red Sea and Suez Canal is projected to increase strongly[55], with a concomitant rise in NO$_x$ emissions. From the increase of NO$_x$ emissions in the model (comparing with and without the RSDW emissions) it is expected that the degassing hydrocarbons will amplify ozone formation in the future (Fig. 5). The photochemical pollution from anthropogenic NO$_x$ and degassing NMHCs from RSDW will directly affect air quality, for example in Neom city, a cross-border megaproject in the Tabuk Province of north-western Saudi Arabia[56].

## Methods

**The AQABA ship campaign.** To study the Air Quality and climate in the Arabian Basin (AQABA), a ship expedition was conducted in July and August 2017. The research vessel *Kommandor Iona* (IMO: 8401999, flag: UK, length overall × breadth extreme: 72.55 m × 14.9 m) was equipped with five air-conditioned laboratory containers that hosted a large suite of atmospheric gas and aerosol measurement equipment. The ship sailed from Toulon (France), crossed the Mediterranean Sea, and through the Suez Canal covered the periphery of the Arabian Peninsula to Kuwait and back. In total, 20,000 km of the marine route was covered with an average speed of 3.4 ± 1.8 m s⁻¹ over the course of 60 days. Further information on the AQABA ship campaign can be found elsewhere in the literature[16,17,57].

**Non-methane hydrocarbon measurements.** Non-methane hydrocarbons (C2–C8) were measured in situ with two coupled, commercial gas chromatography-flame ionization detectors (GC-FID; AMA Instruments GmbH, Germany). Detailed information on the instrumentation, experimental setup, sampling, and calibrations can be found elsewhere[16]. Briefly, atmospheric samples were collected through a common 5.5-m tall (3 m above the container), 0.2-m-diameter, high-flow (≈10 m³ min⁻¹) stack with a subflow of 2.5 L min⁻¹. The air passed through a PTFE filter (5-μm pore size, Sartorius Corporate Administration GmbH, Germany) and heated (40 °C) Teflon lines before it was drawn into the instruments with a flow of 90 sccm (2 × 45 cm³ (stp) min⁻¹ (sccm)). An ozone scrubber (Na₂S₂O₃-infused quartz filters) and a Nafion dryer (500-sccm counter-flow) were used to eliminate the effects of ozone and humidity in sample collection. The sampling times and volumes were adjusted according to ambient NMHC concentrations and wave conditions. During polluted conditions (e.g., Arabian Gulf and the Suez Canal) short sampling times (10 min) and small volumes (450 mL) allowed higher time resolution (50 min per measurement), while under clean conditions, such as those found in the Arabian Sea, longer sampling times and volumes (30 min, 1350 mL, time resolution = 1 h) improved detection limits. For most of the route, the sampling time was 20 min, the sampling volume 900 mL, and the time resolution 50 min per measurement. The diverse conditions met during the campaign led to the geographical demarcation that was used during data analysis (Supplementary Fig. 1).

**Model simulations.** In this work the EMAC (ECHAM5/MESSy Atmospheric Chemistry) model has been used. The EMAC model is a numerical chemistry and climate simulation system that includes submodels describing tropospheric and middle atmosphere processes and their interaction with oceans, land, and human effects[18]. It uses the second version of the Modular Earth Submodel System (MESSy2)[4] to link multi-institutional computer codes. The core atmospheric model is the fifth-generation European Centre Hamburg general circulation model (ECHAM5)[19,58]. For this study, we applied EMAC (ECHAM5 version 5.3.02, MESSy version 2.53.0) in the T106L31 resolution, i.e., with a spherical truncation of T106 (corresponding to a quadratic Gaussian grid of ~1.1 by 1.1° in latitude and longitude) with 31 vertical hybrid pressure levels up to 10 hPa. The simulations cover the period of the AQABA field campaign, i.e., from June to September 2017. The dynamics were weakly nudged by Newtonian relaxation toward ERA-Interim reanalysis data[59]. The model configuration of the chemical mechanism is similar to that of Lelieveld et al.[60], where the comprehensive MOM (Mainz Organic Mechanism)[61] NMHC chemistry representation has been used. Biomass-burning and anthropogenic emissions were prescribed based on the global fire assimilation system[62] and EDGAR, v4.3.2[20,21] database, respectively. Furthermore, the emissions of ethane were subdivided into different source sectors as shown in Supplementary Figs. 2, 5, and 6. The ethane, propane, n-butanes, and i-butane emissions were scaled by factors of 1.9, 1.7, 1.0, and 0.43, respectively, to match recent global emission estimates in the literature[4,15]. Geothermal sources in the region[14] were estimated by scaling sulfuric volcanic emissions to 0.2 Tg yr⁻¹. All emissions were vertically distributed following the literature[58]. Gas flares were estimated based on the work of Caseiro et al.[63].

**Positive matrix factorization.** PMF is a receptor model that uses an advanced multivariate factor analysis technique that is based on weighted least-square fits using realistic error estimates to weight data values, and by imposing non-negativity constraints in the factor computational process. PMF is widely used to identify and quantify the main sources of atmospheric pollutants[64–66]. The mathematical background of PMF analysis is comprehensively described elsewhere[67]. Briefly, the statistical method uses a mass balance equation, which in the receptor model is expressed as

$$X_{ij} = \sum_{k=1}^{p} G_{ik}F_{kj} + E_{ij}, \qquad (1)$$

Here, $X_{ij}$ is the concentration of $j$ species measured in sample $i$ and $G_{ik}$ is the species contribution of the $k$ source to sample $i$. $F_{kj}$ (frequently reported as source profiles) is the fraction of $j$ species from the $k$ source, while $E_{ij}$ is a residual associated with the $j$ species concentration measured in the $i$ sample. Finally, $p$ denotes the total number of the sources. The goal of the model is to reproduce $x_{ij}$ matrix by finding values for $G_{ik}$ and $F_{kj}$ matrices for a given $p$. The values of $G_{ik}$ and $F_{kj}$ matrices are adjusted until a minimum $Q$ (the loss function)

for a given $p$ is found[68]. PMF solves the receptor modeling problem by minimizing the loss function $Q$ based on the uncertainty of each observation by the following equation:

$$Q = \sum_{i=1}^{n} \sum_{j=1}^{m} \left( \frac{e_{ij}}{\sigma_{ij}} \right)^2, \qquad (2)$$

where $\sigma_{ij}$ is an estimate of the uncertainty for the $j$th species in the $i$ sample, $n$ is the number of samples, and $m$ is the number of species.

PMF application to volatile organic compound (VOC) source apportionment and profile contribution has been applied to a wide range of environments including urban and rural areas[69–72]. A main advantage of PMF is that it can provide the source profile and contribution without any prior knowledge of VOC emission profiles. In this study, PMF was applied to the 50-min data samples of the AQABA campaign for identification and quantification of the major observed VOC sources, using the US EPA PMF 5.0 software[22] (https://www.epa.gov/air-research). Missing points were replaced with the median concentration of the corresponding species over the entire measurement matrix and they were accompanied by an uncertainty of 4 times the species-specific median, as suggested[22]. It should be mentioned here that this is the first application of PMF to data from a moving platform (ship). This might introduce a small bias, despite the fact that the data were filtered for own ship exhaust. It should also be noted that the background was not removed from the measurements, since the background changes as the ship travels.

Since PMF is a weighted least-squares method, individual estimates of the uncertainty in each data value are necessary. The uncertainty input data matrix followed established approaches[22,73] by including the measurement uncertainty of each sample and NMHC species[16]. As a complementary criterion, a signal-to-noise condition was additionally applied in the data as suggested in the literature[72]. Individual species that retained a significant signal were separated from those dominated by noise. When signal-to-noise (S/N) ratio was <0.2, species were judged as bad and removed from the analysis. Species with $0.2 < $S/N$ < 2$ were characterized as weak and their uncertainty was tripled. Species with S/N ratio greater than 2 (S/N > 2) were defined as strong and remained unchanged.

As PMF is a descriptive model, there are no objective standards for choosing the right number of factors[66]. However, in order to acquire realistic source profiles and an optimum number of factors, a multicriterion was applied. This included the symmetric distribution of scaled residuals ($\pm 3\sigma$), the investigation of all $Q$ values ($Q_{true}$, $Q_{robust}$, and $Q_{expected}$) (see Eq. (2)), and the interrelationship investigations between the predicted and observed volume mixing ratios. Monitoring of $Q/Q_{exp}$ index with increasing number of factors was used to identify the optimal mathematical solution. The $Q$ value is an assessment of how well the model fits the input data. The difference between the modeled $Q$ value and the theoretical $Q$ value gives a good indication of the suitability of the chosen number of factors. A large decrease in the ratio is indicative of increased explanatory power in the model of the data, while a small decrease is suggestive of little improvement with extra factors. As a consequence, in most areas the number of factors was chosen after $Q/Q_{exp}$ index decreased significantly (Supplementary Fig. 16). In the PMF analysis, the $Q/Q_{exp}$ values represented the ratios between the actual sum of the squares of the scaled residuals ($Q$) obtained from the PMF least-squares fit and the ideal $Q$ ($Q_{exp}$), which was obtained if the fit residuals at each point were equal to the noise specified for each data point. The optimum solution suggested by the $Q/Q_{exp}$ ratio is 4–5 factors, depending on the region of application; however, when the 5-factor solution was examined, a split in the factor was observed. Thus a 4-factor solution was selected. It has to be noted that factor 2 (Fig. 1) NMHC signature remained relatively unchanged under both 4- and 5-factor outputs. Nine different modeling conditions were examined with $p$ values (number of factors) ranging from 2 to 10 where each simulation was randomly conducted 20 times.

To evaluate the reproducibility of the PMF solution and the adequate number of PMF factors with specific focus on the original submatrix $F$, the bootstrap technique was applied[22,73,74]. A bootstrap data set was constructed by sampling blocks of observations from the original data set in random order until reaching the size of the original input data. A base bootstrap model method was carried out, executing 100 iterations, using a random seed and a minimum Pearson correlation coefficient ($R$ value) of 0.6 as suggested in the literature[69,73]. All the modeled factors were well reproduced over at least 85% of runs, indicating that the model uncertainties can be interpreted, and that the number of factors is appropriate. The remaining 15% was distributed among the existing factors, while it should be noted that no runs were unmapped (unmapped is considered a factor when the bootstrap factor is not correlated to any of the base factors). For all the factors, 90% of the species (of the base run) were within the interquartile range (25th–75th percentile) of the bootstrap runs, hence highlighting the robustness of the PMF. The correlation between total VOC-reconstructed concentrations from all four factors with total VOC-observed concentrations is depicted in Supplementary Fig. 19. $R^2$ was 0.92, indicating good agreement between the receptor model and the observations (Supplementary Fig. 18). This also highlights that PMF model explained almost all variance of the total concentration of the 18 VOCs.

## Data availability

The data are available upon request to all scientists agreeing to the AQABA protocol (https://doi.org/10.5281/zenodo.3050041).

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

## Acknowledgements

We acknowledge the fruitful collaborations with the Cyprus Institute (CyI), the King Abdullah University of Science and Technology (KAUST), and the Kuwait Institute for Scientific Research (KISR). We are grateful to Hays Ships Ltd, the ship's captain Pavel Kirzner, and the ship crew for providing the best possible working conditions onboard Kommandor Iona. We thank all the participants of the AQABA ship campaign and in particular Hartwig Harder for the fruitful discussions and day-to-day organization of the campaign, and Marcel Dorf, Claus Koeppel, Thomas Klüpfel, and Rolf Hofmann for logistics organization and assistance during the setup phase. Uwe Parchatka is acknowledged for his contribution to the NO$_x$ data set. Marc Delmotte, Laurence Vialettes, and Olivier Laurent are acknowledged for helping with setting up the methane measurements. NW acknowledges funding by the European Union's Horizon 2020 research and innovation program under the Marie Skłodowska-Curie grant agreement No. 674911. We acknowledge the EMME-CARE project from the European Union's Horizon 2020 Research and Innovation Programme (grant agreement No. 856612), as well as matching co-funding by the Government of the Republic of Cyprus.

## Author contributions

E.B. performed the NMHC measurements, analyzed the data, and drafted the article. A.P. conducted the model simulations. T.S. investigated the potential RSDW sources. VM performed the PMF analysis. L.E. performed the NMHC measurements during the second leg. A.E., C.S., E.P. and N.W. provided the PTR–ToF–MS data, I.T. and H.F. the NO$_x$ data, and J.D.P. the methane concentrations. D.W. and T.K. analyzed the back trajectories. S.O. investigated the circulation in the Red Sea. J.L. conceived and directed the overall project. J.W. supervised the measurements and interpretation. All authors contributed to editing the article.

## Competing interests

The authors declare no competing interests.
