## [Peer Review File · Nature Communications]

Reviewers' comments:

Reviewer #1 (Remarks to the Author):

Manuscript Review

Title: Red Sea deep brine pools - a potent natural source of atmospheric ethane and propane
Submitted by Bourtsoukidis et al.

The investigations described in the submitted manuscript are comparable to the global approach of Dalsoren et al., published in Nature in 2018 ("Discrepancy between simulated and observed ethane and propane levels explained by underestimated fossil emissions"), but focus on the Arabian Peninsula and adjacent waters and identify a yet unconsidered geological source of NMHC into the atmosphere. Assuming that the current emission inventory used in their model is sufficiently well defined - it is mentioned that only few measurements and data are published for the region of interest - a huge emission of ethane and propane is indicated by comparing modelled with measured atmospheric gas concentrations. The calculated ethane+propane emission is comparable to the respective annual emissions from Iraq or UAE.

The authors anticipate that the emission source is located at the seafloor in the Northern Red Sea. Two brine-filled Red Sea deeps in the area (i.e. Kebrit and Oceanographer Deep) are known from literature containing high concentrations of dissolved methane, ethane and propane. However, transport processes from brine-seawater interfaces through 1400 m of water column up to the sea surface are unknown. Density stratification (thermocline), inflow and outflow of water masses (residence times), vertical transport times, and secondary (microbial) degradation of NMHC would play a major role for the local flux of hydrocarbons into the atmosphere.

State of the art onboard gas chromatography was used to measure VOC composition of air along the research track around the Arabian Peninsula. Established statistical approaches were used to identify emission sources in the area (Group/factor analyses). Atmospheric reaction and transport model EMAC was applied to simulate degradation and formation of trace gases and to back-trace the origin of emission. The measured atmospheric pollutant composition, modelled atmospheric hydrocarbon distribution and transport directions, and ozone and PAN formation in the area of interest is a valuable information by itself.

In summary, the manuscript is very well written and the quality of measurements, modelling and presentation of data and interpretation is very good. The topic is of general interest for environmental chemists, oceanographers, geologists, climatologists etc..

I suggest accepting the MS for publication after considering the following concerns (major revisions).

1. Deep-reaching eddies:

Vertical transport of water masses by eddies could be a reasonable approach for the upper 200-300 m water column in the northern Red Sea, as demonstrated by published data (see submitted MS). The normal vertical transport from seafloor to sea surface by (turbulent) mixing is in the range of several 100 years for a water column of about 1000 m. Thus, to convince the oceanographic community about a deep impact into RSDW by eddies one should discuss this a bit more in detail. E.g. eddies can indeed affect deep water environment down to about 4000 m with downward effective transfer rate of 200–600 m·day⁻¹ (e.g. Aleynik et al. 2017).

2. Flux calculations:

Density differences at brine-seawater interfaces of Kebrit and Oceanographer Deep are about 0.2 g/cm³ per 3 m. Transport of constituents is controlled by diffusion. Concentrations of ethane+propane, water depths of interfaces and bathymetry are published. Thus the total area for emissions from brine-seawater interfaces can be estimated. Why not calculating maximum possible

fluxes of NMHC at steady state conditions? When doing a rough calculation of e.g. methane flux across brine +seawater interface at Kebrit Deep, using Fick's first law, my estimated emission is about 390 kg CH₄ per year from Kebrit Deep (see also Tab. 3 in Schmidt et al., 2003). The brine-seawater interface of Oceanographer Deep is of similar size and water depth, compared to Kebrit Deep. So we are talking about a flux of less than 1x10⁶ g CH₄ per year from these two brine pools. This is several orders of magnitude lower than the predicted 1.3 Tg CH₄ per year emission rate from Kebrit and Oceanographer brines, calculated in the submitted manuscript. Obviously there is a mismatch and I suggest to consider additional sources for methane and NMHC from depth.

3. Additional methane and NMHC sources:

Additional hydrocarbon release from seafloor is possible due to fluid seepage from a partly fractured seafloor deposited above offshore source rocks, located between 25 and 28°N (e.g. Rudeis and Kareen formation).

Upwelling of hydrocarbon-enriched bottom water derived from inflow waters originated in the Gulf of Suez, e.g. crude oil/gas seepage from leaky subsea wells could be an additional source. The outflow of dense hydrocarbon-rich water from Gulf of Suez feeding the RSDW and subsequent upwelling in the Northern Red Sea is already mentioned in the manuscript.

The upwelling concept is mainly derived from tracer studies and the respective paper should be cited (Jean-Baptiste et al., 2004). Consider the full range of estimated residence times for RSDW:

~10 years (Yao and Hoteit, 2018)

~26-60 years (Jean-Baptiste et al., 2004)

4. Temporal variability of emissions (triggered events):

One should think about triggered emission events e.g. by seismic events in this area. Earth quake (M4.4) in central graben 144 km northwest of Jeddah on June 19th, 2017 fueling the RSDW with oil-associated gas? This was only one month before the measuring campaign started. Maybe the high ethane and propane anomalies in the Northern Red Sea is a temporal phenomenon? This would change the estimated annual flux calculation significantly.

5. Secondary microbial oxidation

Lines 179-181: you are citing anaerobe microbial oxidation here. However, degradation of hydrocarbons in the water column mainly proceeds by aerobic oxidation. So I don't think 42 and 43 are appropriate here. Moreover, ethane and propane are effectively oxidized in an oxygenated water column and bacterial degraders preferentially oxidize ethane (e.g. Valentine et al., 2010).

Minor corrections:

Line 496: Give full citation of reference "20"

Legend in Fig.1-d: correct "Measuremenets" and increase symbol size (dot)

Added references:

Valentine, et al. (2010) Propane Respiration Jump-Starts Microbial Response to a Deep Oil Spill. *Science* 330, 208. DOI: 10.1126/science.1196830

Aleynik et al. (2017) Impact of remotely generated eddies on plume dispersion at abyssal mining sites in the Pacific. *Sci. Rep.* 7 (1), p. 16959, 10.1038/s41598-017-16912-2

Jean-Baptiste et al. (2004) Red Sea deep water circulation and ventilation rate deduced from the He-3 and C-14 tracer fields. *J. Mar. Syst.* 48, 37-50.

M4.4 earth quake:

https://www.researchgate.net/publication/257786054_A_spatial_statistical_analysis_of_the_occurrence_of_earthquakes_along_the_Red_Sea_floor_spreading_Clusters_of_seismicity/figures

sign. Mark Schmidt

Reviewer #2 (Remarks to the Author):

This paper identifies a source of ethane and propane to the atmosphere previously unrecognized and not included in standard emissions inventories, namely from brine pools in the Red Sea. Ethane and propane are important precursors of air pollution (ozone), and have been generally underestimated in model simulations due to an incomplete understanding and quantification of their sources. These results are important for improving prediction of air quality in the region, and may have increasing importance as the types of emissions sources in the region (such as shipping) evolve.

The paper is well written and clearly organized. The methods are explained in sufficient detail, in particular the positive matrix factorization used to determine the strength of the new emissions source is clearly described. The figures clearly illustrate the results.

I recommend publication of this paper and have just a few minor suggestions:

1) I recommend plotting the ratio of model / measurement, instead of meas./model as you do in the main text and Supplement. The way they are plotted, I think the implication is that the measurements are too high, but your point really is that the model is too low. I think this would be more clearly shown if the measurements (truth) are the reference (model/meas.), as the scatter plots show, having the measurements on the x-axis. However, if there is a good argument for showing (meas./model) it is not necessary to change it.

2) Ref. 20 is incomplete.

3) Supp. Figures 14-16: Explain what the max value is exactly – maximum of daily, hourly values in the 3 months?

4) It might be nice to have a figure in the main text illustrating the implications of this missing source. Perhaps Supp. Fig. 17 could be included in the main text?

5) Another reference that illustrates the impact of correcting ethane emissions on PAN and O₃ distributions is Monks et al., JGR, <https://doi.org/10.1029/2017JD028112>, 2018.

Dear Dr. Frischkorn,

Dear reviewers,

Thank you very much for reviewing our work and for providing the insightful and helpful comments. After reflecting on these comments we agree that we cannot unequivocally identify the deep brine pools as the single missing submarine source of hydrocarbons in the Red Sea. Although for the reasons now discussed in the manuscript, the brine pools remain among the potential sources, we now include all possible deep sea sources of NMHCs (and CH₄) from the Red Sea Deep Water (RSDW). The title of the paper is revised accordingly. The main point of the paper still stands, that beneath the Red Sea there is a very important source of hydrocarbon gases (particularly ethane and propane) which enters the atmosphere but are missing in current inventories, with substantial implications for the regional atmospheric chemistry.

We are grateful to both reviewers who have acknowledged the quality of our measurements, modelling and presentation of data and interpretation in the first interaction. We believe that the revised manuscript comprehensively addresses all concerns raised.

Please note that in the revised version, we have included Dr. Sergey Osipov (Max Planck Institute for Chemistry) who has research experience on the field and has further investigated the water circulation in the Red Sea for the purposes of this manuscript.

Below, we provide detailed responses to the reviewers' comments (in blue) and denote the changes in the text of the revised manuscript (in green). All revisions performed are additionally provided in the attached document that tracks the changes in the text.

Reviewer #1 (Remarks to the Author):

Manuscript Review

Title: Red Sea deep brine pools - a potent natural source of atmospheric ethane and propane

Submitted by Bourtsoukidis et al.

The investigations described in the submitted manuscript are comparable to the global approach of Dalsoren et al., published in Nature in 2018 ("Discrepancy between simulated and observed ethane and propane levels explained by underestimated fossil emissions"), but focus on the Arabian Peninsula and adjacent waters and identify a yet unconsidered geological source of NMHC into the atmosphere. Assuming that the current emission inventory used in their model is sufficiently well defined - it is mentioned that only few measurements and data are published for the region of interest - a huge emission of ethane and propane is indicated by comparing modelled with measured atmospheric gas concentrations. The calculated ethane+propane emission is comparable to the respective annual emissions from Iraq or UAE.

The authors anticipate that the emission source is located at the seafloor in the Northern Red Sea. Two brine-filled Red Sea deeps in the area (i.e. Kebrit and Oceanographer Deep) are known from literature containing high concentrations of dissolved methane, ethane and propane. However, transport processes from brine-seawater interfaces through 1400 m of water column up to the sea surface are unknown. Density stratification (thermocline), inflow and outflow of water masses (residence times), vertical transport times, and secondary (microbial) degradation of NMHC would play a major role for the local flux of hydrocarbons into the atmosphere.

State of the art onboard gas chromatography was used to measure VOC composition of air along the research track around the Arabian Peninsula. Established statistical approaches were used to identify emission sources in the area (Group/factor analyses). Atmospheric reaction and transport model EMAC was applied to simulate degradation and formation of trace gases and to back-trace the origin of emission. The measured atmospheric pollutant composition, modelled atmospheric hydrocarbon distribution and transport directions, and ozone and PAN formation in the area of interest is a valuable information by itself.

In summary, the manuscript is very well written and the quality of measurements, modelling and presentation of data and interpretation is very good. The topic is of general interest for environmental chemists, oceanographers, geologists, climatologists etc..

I suggest accepting the MS for publication after considering the following concerns (major revisions).

We are very grateful for the insightful and detailed comments provided by Dr. Schmidt. In particular, he has drawn our attention to alternative deep sea hydrocarbon sources in the region and noted the uncertainty in our identification of the brine pool source. In order to take this information into account, we have now revised the title of the manuscript to *“Red Sea Deep Water – a potent source of atmospheric ethane and propane”* and thoroughly discuss the concerns raised.

1. Deep-reaching eddies:

Vertical transport of water masses by eddies could be a reasonable approach for the upper 200-300 m water column in the northern Red Sea, as demonstrated by published data (see submitted MS). The normal vertical transport from seafloor to sea surface by (turbulent) mixing is in the range of several 100 years for a water column of about 1000 m. Thus, to convince the oceanographic community about a deep impact into RSDW by eddies one should discuss this a bit more in detail. E.g. eddies can indeed affect deep water environment down to about 4000 m with downward effective transfer rate of 200–600 m·day⁻¹ (e.g. Aleynik et al. 2017).

Thank you for pointing out the mentioned study and the insights on eddies' impact over the deep water circulation dynamics. We incorporated a thorough discussion of the vertical transport mechanisms into the manuscript. This includes upwelling in the northernmost Red Sea as a part of the complex three-dimensional overturning circulation and pumping of the RSDW by the eddies. We also highlight in the manuscript that, based on the seasonal and multiyear variability of the vertical transport, the real emissions could be significantly amplified, and our estimate therefore represents the lower boundary of the emissions. This information and respective literature has been now added to the manuscript.

Added text: Further, mesoscale eddies that are particularly effective in the areas of the Oceanographer and Kebrit deeps and may contribute to the vertical transport of hydrocarbons. Eddies may also affect deep water environment with downward effective transfer rate of 200–600 m·day⁻¹ as measured in the Pacific Ocean.

2. Flux calculations:

Density differences at brine-seawater interfaces of Kebrit and Oceanographer Deeps are about 0.2 g/cm³ per 3 m. Transport of constituents is controlled by diffusion. Concentrations of ethane+propane, water depths of interfaces and bathymetry are published. Thus the total area for emissions from brine-seawater interfaces can be estimated. Why not calculating maximum possible fluxes of NMHC at steady state conditions? When doing a rough calculation of e.g. methane flux across brine +seawater interface at Kebrit Deep, using Fick's first law, my estimated emission is about 390 kg CH₄ per year from Kebrit Deep (see also Tab. 3 in Schmidt et al., 2003). The brine-seawater interface of Oceanographer Deep is of similar size and water depth, compared to Kebrit Deep. So we are talking about a flux of less than 1x10⁶ g CH₄ per year from these two brine pools. This is several orders of magnitude lower than the predicted 1.3 Tg CH₄ per year emission rate from Kebrit and Oceanographer brines, calculated in the submitted manuscript. Obviously there is a mismatch and I suggest to consider additional sources for methane and NMHC from depth.

We agree that considering diffusion rates alone will result in orders of magnitude smaller fluxes for ethane, propane and for the example mentioned above methane. Indeed, applying the diffusion calculations alone, the emissions from the brine pools are very small and unable to explain the high concentrations that were measured during our campaign.

In general agreement with this comment, we now also add that as the above-brine water currents remain unquantified, and the diffusion of hydrocarbons may be further enhanced by deep sea currents traversing the brines. We also discuss the renewal of the RSDW and point out that due to the weak column stratification, the open-ocean convection and vertical mixing in the northernmost Red Sea can be especially vigorous during the winter seasons and during the positive phase of the Arctic Oscillation. For example, Genin et al., 1995 observed unusually deep mixing extending to >850 m. The instabilities in the water column can produce rapid vertical transport, which can reach the seafloor and enhance the uptake of the hydrocarbons from the brine pools. Nonetheless, it is correctly pointed out that further sources should be considered (see response at point 3). Hence, we now refer to the literature suggested on the weak diffusion rates that seems to be the single driver in the case of Kebrit deep (Schmidt et al., 2003). Additionally, we have now removed the terms "brine pools" and "natural" from the title and discuss all possible sources in a separate section, approaching the measured degassing rates in more holistic approach that considers all points raised.

Added text: *The mixing between the dense brine pools and RSDW occurs via diffusion across the strong salinity gradient and reported methane emissions are relatively weak with values up to 393 Kg/yr for the Kebrit deep. However, the impact of above-brine water currents which may enhance the fluxes has not been quantified, so this emission estimate remains uncertain.*

3. Additional methane and NMHC sources:

Additional hydrocarbon release from seafloor is possible due to fluid seepage from a partly fractured seafloor deposited above offshore source rocks, located between 25 and 28°N (e.g. Rudeis and Kareen formation).

Upwelling of hydrocarbon-enriched bottom water derived from inflow waters originated in the Gulf of Suez, e.g. crude oil/gas seepage from leaky subsea wells could be an additional source. The outflow of dense hydrocarbon-rich water from Gulf of Suez feeding the RSDW and subsequent upwelling in the Northern Red Sea is already mentioned in the manuscript.

The upwelling concept is mainly derived from tracer studies and the respective paper should be cited (Jean-Baptiste et al., 2004). Consider the full range of estimated residence times for RSDW:

~10 years (Yao and Hoteit, 2018)

~26-60 years (Jean-Baptiste et al., 2004)

Thank you very much for this very insightful comment. Indeed, fluid seepage and crude oil/gas seepage from leaky subsea wells seem to be very likely sources. Therefore, we now emphasize the importance of these sources and discuss with appropriate references.

Added text on the emissions: *Hydrocarbon release from the Red Sea floor can occur through direct fluid seepage from deposits above offshore rocks, located between 25 and 28°N (e.g. Rudeis and Kareen formation). In addition, the Gulfs of Suez and Aqaba contribute to the RSDW through bottom-trapped density outflows. Considering that this region is known for the large oil and gas reserves, natural seepage and crude oil/gas seepage from leaky subsea wells could be a significant submarine source of NMHCs in this region. Finally, the numerous brine pools that are located on the seafloor need to be considered.*

And

Considering the aforementioned potential sources, it seems entirely plausible that the RSDW is highly enriched in ethane and propane from a deep sea source. The transfer from deep water to the surface can be relatively rapid due to the exceptionally effective vertical mixing of the Red Sea's deep water and the outflows from the Gulfs of Suez and Aqaba that have been considered to be important for the RSDW renewal in the period 1982-2001.

Added text on the upwelling: *The upwelling of the intermediate and deep water takes place in the narrow band along the Egyptian coast. Spatially, the upwelling is restricted to the northmost Red Sea (north of 24°N) and coincides with the location of the missing NMHC source. The complex overturning circulation in the Red Sea has a pronounced seasonal cycle that modulates the vertical transport and can potentially amplify the emissions to the atmosphere during winter. The upwelling is weaker in summer compared to the winter since atmospheric cooling drives the open water convection and enhances the vertical mixing in the water column. Due to the weak stratification of the water column in the northern part of the basin, the convective mixing can be especially deep and reach the sea floor. Model estimates of the renewal times range from 19 to 90 years, while tracer studies indicate somewhat faster renewal times to about 26 years. Further, mesoscale eddies that are particularly effective in the areas of the Oceanographer and Kebrit deeps and may contribute to the vertical transport of hydrocarbons. Eddies may also affect deep water environment with downward effective transfer rate of 200–600 m·day⁻¹ as measured in the Pacific Ocean.*

While the relative significance of the various submarine hydrocarbon sources cannot be ascertained, we assume that their cumulative contribution represents the missing source derived in this study. This assumption is supported by the similarity in the PMF derived chemical emission profiles (increased alkane concentrations and the absence of alkenes and other anthropogenic tracers in the emission signature). Therefore, we surmise that methane and non-methane hydrocarbons can potentially reach the surface and degas into the atmosphere following air-sea exchange mechanisms.

Added sentence in the Implications section:

Considering the seasonality of the deep water circulation, it is likely that the emissions to the atmosphere will be further enhanced during the wintertime.

4. Temporal variability of emissions (triggered events):

One should think about triggered emission events e.g. by seismic events in this area. Earth quake (M4.4) in central graben 144 km northwest of Jeddah on June 19th, 2017 fueling the RSDW with oil-

associated gas? This was only one month before the measuring campaign started. Maybe the high ethane and propane anomalies in the Northern Red Sea is a temporal phenomenon? This would change the estimated annual flux calculation significantly.

Triggered emissions from seismic events may indeed alter the temporal variability of the degassing hydrocarbons. However, quantifying such alterations would require continuous monitoring of atmospheric hydrocarbons that cannot be adequately addressed with the available data from the region. Considering our measurements, we did not observe a significant difference between the two crossings over July and August (ca. 0.5 and 2 months after the earthquake) as the average meas./model ratios for the referred area is 4.3 ± 4.1 for Leg 1 (median = 3.1) and 4 ± 2.5 for Leg 2 (median = 3). Please note here that the lower standard deviation of Leg 2 is the result of a more rapid crossing that contains ca. 1/3 of the data collected in the first leg ($N_{\text{leg1}} = 159$, $N_{\text{leg2}} = 51$).

The suggested paper by Al-Ahmadi et al. (2014) reports on 553 seismic events of magnitude >4 and an astonishing sum of 1888 seismic events of magnitude >3 over a time frame of 109 years for the complete Red Sea. While the intensity differentiates between the northern, middle and southern part of Red Sea, seismicity over the area is clearly a very common phenomenon. We appreciate the validity of this argument as the earthquake was rather more intense than the norm so we now add to the discussion the potential earthquake triggered emissions in this region.

Added text: *Increasing uncertainties will be introduced when considering the seasonality of the deep-water circulation and, the wind speeds at the air-sea interface and the potential temporal variability of the emissions (i.e. triggered events by the increased seismic activity in the region).*

5. Secondary microbial oxidation

Lines 179-181: you are citing anaerobe microbial oxidation here. However, degradation of hydrocarbons in the water column mainly proceeds by aerobic oxidation. So I don't think 42 and 43 are appropriate here. Moreover, ethane and propane are effectively oxidized in an oxygenated water column and bacterial degraders preferentially oxidize ethane (e.g. Valentine et al., 2010).

This is another valuable comment which helps improve the description of the in-sea processes. We have now revised the respective literature as suggested

Changes made: In addition to the suggested study, we also refer to the review study by Antuenes et al. (2011).

Minor corrections:

Line 496: Give full citation of reference "20"

Thank you for noticing this. The full citation is now provided (Ref. XX in the revised version).

Legend in Fig.1-d: correct "Measuremenets" and increase symbol size (dot)

Thank you for noticing this typo. It has been now corrected.

Added references:

Valentine, et al. (2010) Propane Respiration Jump-Starts Microbial Response to a Deep Oil Spill. Science 330, 208. DOI: 10.1126/science.1196830

Aleynik et al. (2017) Impact of remotely generated eddies on plume dispersion at abyssal mining sites in the Pacific. Sci. Rep. 7 (1), p. 16959, 10.1038/s41598-017-16912-2

Jean-Baptiste et al. (2004) Red Sea deep water circulation and ventilation rate deduced from the He-3 and C-14 tracer fields. J. Mar. Syst. 48, 37–50.

M4.4 earth quake:

https://www.researchgate.net/publication/257786054_A_spatial_statistical_analysis_of_the_occurrence_of_earthquakes_along_the_Red_Sea_floor_spreading_Clusters_of_seismicity/figures

sign. Mark Schmidt

Concluding remark: Dr. Schmidt has submitted a very good and meticulous review. We highly appreciate his remarks and directions towards a comprehensive discussion over the potential hydrocarbon sources. His insights, have helped us to revise the manuscript and main claim (i.e. that brine pools are the single source of the observed emissions). Therefore, we would like to acknowledge his contribution in the respective section as follows:

“We acknowledge the insightful reviewers and are grateful to Dr. Mark Schmidt (GEOMAR – Helmholtz Center for Ocean Research Kiel) who provided valuable suggestions regarding the RSDW sources and oceanic processes discussed in this article. “

Despite the revised and additional discussion of the underwater sources, our flux calculations remain unchanged due to the known constraints of the EMAC grid resolution. As mentioned in the methods and new text, the two point sources that have been added to the EDGAR emission inventory are ca. 2 * 122km² (1.1 x 1.1 degrees) and cover a large surface of the Northern Red Sea, including fluid seepage from the partially fractured seafloor and hydrocarbon enriched density outflows from the Gulfs of Suez and Aqaba. In other words, the global atmospheric model used in this study does not have the resolution to differentiate between the deep sea sources. The main point of the paper still stands, that beneath the Red Sea at the location of the model’s gridbox, there is a very important source of hydrocarbon gases (particularly ethane and propane) which enters the atmosphere. We no longer identify this as the brine pools but instead discuss more thoroughly the diverse sources suggested by the reviewer.

Reviewer #2 (Remarks to the Author):

This paper identifies a source of ethane and propane to the atmosphere previously unrecognized and not included in standard emissions inventories, namely from brine pools in the Red Sea. Ethane and propane are important precursors of air pollution (ozone), and have been generally underestimated in model simulations due to an incomplete understanding and quantification of their sources. These results are important for improving prediction of air quality in the region, and may have increasing importance as the types of emissions sources in the region (such as shipping) evolve.

The paper is well written and clearly organized. The methods are explained in sufficient detail, in particular the positive matrix factorization used to determine the strength of the new emissions source is clearly described. The figures clearly illustrate the results.

I recommend publication of this paper and have just a few minor suggestions:

We thank the reviewer#2 for recognizing the importance of our discovery and the respective atmospheric chemistry implications of the degassing ethane and propane. We particularly appreciate the feedback on the increasing importance of these emissions and now include the former Supplementary Figure 17 in the main text.

1) I recommend plotting the ratio of model / measurement, instead of meas./model as you do in the main text and Supplement. The way they are plotted, I think the implication is that the measurements are too high, but your point really is that the model is too low. I think this would be more clearly shown if the measurements (truth) are the reference (model/meas.), as the scatter plots show, having the measurements on the x-axis. However, if there is a good argument for showing (meas./model) it is not necessary to change it.

We appreciate the suggestion to reverse the ratio between the measurements and the model that would in turn emphasize the model underestimation. However, we would prefer to keep the measurements in the numerator since the uniqueness of our finding lies on the extremely high ambient concentrations measured. We feel that the current approach is better aligned with the title of our study where we report “a potent source of atmospheric ethane and propane”. By plotting meas./model ratio, the discovered “hot spot” is clearly shown in red, a color that commonly associated with intensity and strength. Reversing the order would emphasize a weakness (in the emission inventories) and reduce the clarity regarding the importance of the source. It may be more appealing to the reader a report over a “discovery of a new, potent source” rather than the fact we were ignoring it so far. Nonetheless, we recognize that this is a minor comment and we leave it in the discretion of the editor.

2) Ref. 20 is incomplete.

Thank you for noticing this. The full citation is now provided (Ref. XX in the revised version).

3) Supp. Figures 14-16: Explain what the max value is exactly – maximum of daily, hourly values in the 3 months?

We agree that further information is necessary. We added the following text in the legend of the figures:

“The mean fractional deviations (%) illustrates the spatial average differences (resolution of 1 hour) over the three-month simulations (June, July, August) using the model with and without the RSDW sources. The maximum (minimum for OH) differences illustrate the most relatively largest hourly differences between the two model simulations (original EDGAR v.4.3.2 minus the revised emission inventory)”

4) It might be nice to have a figure in the main text illustrating the implications of this missing source. Perhaps Supp. Fig. 17 could be included in the main text?

We appreciate the suggestion to include the Supp. Fig. 17 in the main text. We agree that by illustrating the implications, the impact of our discovery will become clearer to the reader. Therefore, Supp. Fig. 17 is now Fig. 5 of the main text.

5) Another reference that illustrates the impact of correcting ethane emissions on PAN and O₃ distributions is Monks et al., JGR, <https://doi.org/10.1029/2017JD028112>, 2018.

Thank you for the highly relevant publication. This study is now cited in the appropriate part of the text.

REVIEWERS' COMMENTS:

Reviewer #1 (Remarks to the Author):

Comment on revised version of NCOMMS-19-32149A

All comments and questions are considered in the revised version of the manuscript and changes made improved clarity and plausibility of the message. The new title „Red Sea Deep Water - a potent natural source of atmospheric ethane and propane” is appropriate and reflects the slightly changed focus, concerning different subsea hydrocarbon sources.

I suggest to accept the revised manuscript for publication. See minor correction below:

L146 change to: ...from hydrocarbon reservoirs deposited above offshore source rocks...(e.g....Kareem)...

I enjoyed reading this contribution,
Mark Schmidt

Reviewer #2 (Remarks to the Author):

The authors have addressed my suggestions for improving the manuscript. I recommend publication.